# Differential cognitive and clinical improvements in Schizophrenia and bipolar disorder following hospitalization: A comparative analysis based on the Clock Drawing Test

Ahmad Pirani[1‡], Atiye Sarabi-Jamab[2,3‡], Faeze Bostan[1], Razieh Salehian[1*],
Mehran Ilaghi[4], Ali Aledavoud[5], Seyed Vahid Shariat[1], Fatemeh Sadat Mirfazeli[6,1,7*]

1 Mental Health Research Center, Psychosocial Health Research Institute, Department of Psychiatry, School of Medicine, Iran University of Medical Sciences, Tehran, Iran, 2 Faculty of Governance, University of Tehran, Tehran, Iran, 3 School of Cognitive Sciences, Institute for Research in Fundamental Sciences (IPM), Tehran, Iran, 4 Institute of Neuropharmacology, Kerman Neuroscience Research Center, Kerman University of Medical Sciences, Kerman, Iran, 5 School of Medicine, Kerman University of Medical Sciences, Kerman, Iran, 6 National Brain Centre, Iran University of Medical Sciences, Tehran, Iran, 7 Faculty of Advanced Technologies in Medicine, Iran University of Medical Sciences, Tehran, Iran

‡ Ahmad Pirani and Atiye Sarabi-Jamab contributed equally as first authors.
* mirfazeli.f@iums.ac.ir (FSM); Salehian.r@iums.ac.ir (RS)

## Abstract

### Background

Schizophrenia (SCZ) and bipolar disorder with psychotic features (BDP) are psychiatric disorders with significant impact on affected individuals. However, research comparing cognitive impairments between these groups is limited. This study aimed to evaluate the changes in cognitive function based on the Clock Drawing Test (CDT) among hospitalized SCZ and BDP patients and its association with positive and negative symptoms.

### Methods

This cross-sectional study enrolled 84 Iranian patients (42 SCZ, 42 BDP, 51 male and 33 female) aged 42.85 (±11.51), ranging from 20 to 65 years old, from two psychiatric hospitals in Iran. The Positive and Negative Syndrome Scale (PANSS), Clinical Global Impression (CGI) scale and CDT were administered during the admission and discharge of patients. Within-group and between-group changes were analyzed using paired t-tests for pre-post comparisons and multiple regression was used to assess predictive factors of changes in cognitive and symptom changes.

### Results

Both groups showed cognitive and clinical improvements at discharge, but changes were more pronounced in the BDP group for CDT, PANSS-positive symptoms, and

**Data availability statement:** All relevant data are within the manuscript and its Supporting information files.

**Funding:** The author(s) received no specific funding for this work.

**Competing interests:** The authors declare that no competing and financial interests exist.

**Abbreviations:** SCZ, Schizophrenia; BD, Bipolar Disorder; BDP, Bipolar Disorder with Psychotic features; CDT, Clock Drawing Test; DSM-5, Diagnostic and Statistical Manual of Mental Disorders 5; CNS, Central Nervous System; PANSS, Positive and Negative Scale Symptoms; CGI, Clinical Global Impression; SD, Standard Deviation.

CGI. However, PANSS-negative symptoms improved more in SCZ patients. No correlations existed between changes in CDT and positive and negative symptoms in either of the groups.

## Conclusion

While both groups exhibited cognitive and clinical improvements following hospitalization, patients with SCZ showed relatively less improvement in the CDT compared to those with BDP. These findings suggest that cognitive recovery may follow a different trajectory in SCZ, independent of changes in positive and negative symptoms but related to the initial cognitive profile. However, given that the CDT is a screening tool rather than a comprehensive cognitive assessment, these results should be interpreted with caution. Future research should incorporate broader neurocognitive assessments to better understand the cognitive trajectories of these populations.

## Introduction

Schizophrenia (SCZ) and bipolar disorder (BD) are significant psychiatric disorders that impact a substantial portion of individuals worldwide. Both disorders have similar prevalence rates across genders, with SCZ affecting 0.32% and BD affecting 1% of the global population [1–3]. Generally, there is a wide range of negative and positive symptoms associated with both disorders, with negative symptoms including passive social withdrawal, emotional withdrawal, stereotyped thinking, and motor retardation, and positive symptoms encompassing grandiosity, delusions, a lack of judgment and insight, and hallucinatory behavior [4].

SCZ is often characterized by the presence of delusions, hallucinations, disorganized speech, grossly disorganized or catatonic behavior, as well as negative symptoms [3]. BD, on the other hand, manifests by extreme mood fluctuations, including manic or major depressive episodes, and a propensity for periods of remission and recurrence, which is categorized into three primary types: bipolar type I, II, and cyclothymic disorder [3]. Unlike the second type, the first type exhibits more severe symptoms during the manic phase, which can restrict patient functionality. Similar to SCZ, patients of the first type may exhibit psychotic symptoms, commonly referred to as bipolar disorder with psychotic features (BDP) [5].

Besides the common symptoms observed in SCZ and BDP patients, cognitive deficits also exist in both disorders [6,7]. These manifestations occur as impairments in various domains, including attention, vigilance, working memory, verbal learning and memory, visual learning and memory, speed of processing, reasoning, problem-solving, and social cognition [7–9]. These deficits lead to significant detrimental clinical and social outcomes as well as dependence on others [10]. Therefore, both disorders raise the potential for cognitive interventions as effective treatments. However, while numerous studies have explored the cognitive deficits in SCZ, research on cognitive deficits in BD is relatively scarce, and it is still uncertain whether

enhancements in both positive and negative symptoms, along with neurocognition, happen concurrently or if these progressions are unrelated [11,12].

Treatment approaches for SCZ and BDP typically involve a combination of pharmacotherapy, psychotherapy, and, in some cases, somatic interventions like electroconvulsive therapy (ECT), aimed at reducing symptom severity and improving functional outcomes. For SCZ, antipsychotic medications are the cornerstone of treatment, effectively reducing positive symptoms such as hallucinations and delusions within weeks, though negative symptoms and cognitive deficits often show limited improvement [13]. A meta-analysis by Leucht et al. (2017) found that antipsychotics yield moderate effect sizes (Cohen's d ≈ 0.5) for positive symptom reduction, but cognitive gains are less consistent, with some studies reporting minor improvements in processing speed and attention following second-generation antipsychotics [14]. In BDP, mood stabilizers (e.g., lithium, valproate) combined with antipsychotics during psychotic manic episodes are standard, with evidence suggesting robust reductions in positive symptoms and mania-related cognitive disruptions within 4–6 weeks. A review by Nierenberg et al. (2023) highlights that bipolar patients often experience partial cognitive recovery post-treatment, particularly in executive functioning, though deficits may persist during euthymic phases [5]. These differential treatment responses underscore the need to evaluate how clinical and cognitive outcomes align in hospitalized patients, providing a foundation for understanding potential therapeutic impacts in this study's context.

In the evaluation of patients suffering from cognitive deficits, the selection of a valid and reliable test is crucial in clinical practice. One such test is the Clock Drawing Test (CDT), which was initially employed as a *"parietal lobe"* test, while contemporary research primarily concentrates on its sensitivity and specificity in identifying and distinguishing between different types of dementia [15]. It is projected that the multifaceted cognitive processes underlying the CDT's performance contribute to its high sensitivity in detecting the widespread disruption of brain systems that characterize various forms of neurocognitive disorders. Presently, the application of the CDT has broadened to include screening for cognitive impairments in SCZ patients [12,16].

Despite existing literature, there remains a scarcity of studies that evaluate cognitive deficit recovery and the association between the changes in positive and negative symptoms with alterations in cognitive deficit in both SCZ and BDP patients. In this study, we aimed to evaluate the symptoms and the extent of cognitive impairment in SCZ and BDP patients before and after treatment. We further compared the changes in clinical trajectories between the two disease entities and assessed whether the changes in function were correlated with positive and negative symptoms. We hypothesized that: i) both SCZ and BDP patients would exhibit significant improvements in cognitive function (measured by CDT) and clinical symptoms (measured by Positive and Negative Syndrome Scale (PANSS) and Clinical Global Impression (CGI)) following hospitalization, given the expected efficacy of standard treatments; (2) BDP patients would demonstrate greater improvements in CDT scores and positive symptom reduction compared to SCZ patients, reflecting evidence of less severe baseline cognitive deficits and stronger treatment response in BD; and (3) changes in cognitive function would not strongly correlate with changes in positive or negative symptoms in either group, suggesting that cognitive deficits may represent a distinct domain from psychotic symptomatology.

## Methods

### Study design and participants

This was a cross-sectional study on SCZ and BDP patients hospitalized in two hospitals (Rasoul Akram Hospital and Iran Psychiatry Hospital) located in Tehran, Iran, between September 23, 2021, and December 1, 2022. Inclusion criteria were an age of 18 and above, Farsi speaking and Iranian population, a diagnosis of SCZ or BDP according to the Diagnostic and Statistical Manual of Mental Disorders 5 (DSM-5), and consent to participate in the study. The diagnosis was made by a psychiatrist attending using a comprehensive psychiatric interview. The exclusion criteria included a history of traumatic brain injury, seizure within the last two years, brain tumor, and central nervous system (CNS) diseases such as multiple sclerosis. Moreover, patients who were discharged against medical advice without completing the treatment course were

excluded. Patients were enrolled using convenience sampling, allowing any hospitalized patient meeting the study criteria to participate. There were no dropouts throughout the study. The sample size was determined using G*Power 3.1.9.4 software, based on effect size estimates from a previous study which examined cognitive performance using the CDT in SCZ and Alzheimer's disease (AD) [17]. In their study, a total sample of 100 participants (32 SCZ, 32 Alzheimer's, 36 controls) revealed significant group differences across various clock conditions (F (2,97) = 26.04 to 32.43, p < 0.001). Based on these findings, the estimated effect size (Cohen's d) for cognitive group differences was 0.4. Given our study design, where each participant is assessed at two-time points (Admission and Discharge) for multiple measures (CDT, PANSS Positive, PANSS Negative, and CGI), a paired t-test or repeated-measures ANOVA is appropriate. Using G*Power, we calculated that with α = 0.05, power = 0.95, effect size = 0.4, and two repeated measurements, a total sample size of 64 participants is required to detect significant effects. The achieved statistical power (0.953) ensures adequate sensitivity for detecting clinically meaningful changes.

## Procedures

Demographic data and medication details of the patients were obtained from the patient's medical records. The two study groups (SCZ or BDP) were interviewed by a trained psychiatric specialist at the time of admission and at the end of hospitalization. Patients were assessed at time points using the PANSS, the CGI, and the CDT. The scores obtained in each assessment were recorded at the time of admission and at the time of discharge.

## Instruments

### a. Positive and Negative Scale Symptoms (PANSS)

The PANSS, developed by Kay, Opler, and Fiszbein in 1987, is a widely used psychometric tool for assessing the severity of psychosis symptoms in SCZ and BDP [4,18]. It consists of 30 items divided into three subscales: positive (7 items), negative (7 items), and general psychopathology (16 items). In this study, we focused on the positive (P-PANSS) and negative (N-PANSS) subscales. Each item is scored on a 7-point scale (1 = absent, 7 = extreme), yielding a total score range of 7–49 for each subscale, with higher scores indicating greater symptom severity. The PANSS has demonstrated strong psychometric properties, with high internal consistency (Cronbach's α = 0.73–0.83 for subscales) and test-retest reliability (r = 0.77–0.89) in SCZ populations [18]. Its construct validity is supported by factor analyses confirming the three-subscale structure, and external validity is evidenced by correlations with other clinical measures like the Brief Psychiatric Rating Scale (BPRS) (r = 0.64–0.81) [18,19].

### b. Clinical Global Impression (CGI)

The CGI is a clinician-rated tool used to assess the severity and change in a patient's condition, commonly applied in psychiatric research and clinical practice [20]. In this study, we used the CGI-Severity (CGI-S) subscale, which rates illness severity on a 7-point scale (1 = normal, not at all ill; 7 = among the most extremely ill), with a score range of 1–7; higher scores indicate greater severity. The CGI has shown adequate reliability, with inter-rater reliability coefficients ranging from 0.71 to 0.8 in psychiatric populations [20,21]. Its external validity is evidenced by moderate correlations with symptom-specific scales like the PANSS (r = 0.36 to 0.61) [21]. However, the CGI's subjective nature can introduce variability, which we mitigated by using trained psychiatric specialists for assessments.

### c. Clock Drawing Test (CDT)

The CDT has been widely used as a measure of cognitive function in several disorders since the 1960s. CDT has been linked to dysfunction in the parietal lobe, as well as various brain regions such as the left and right posterior and middle temporal lobe, the right middle frontal gyrus, and the right occipital lobes. In this research, participants were provided with

a white A4 paper and instructed to draw a clock, setting the time to 2:45. Each clock was quantitatively scored, with the scoring checked by a neuropsychiatrist. We utilized the scoring procedure advocated by Mendes-Santos et al., in which the scores range from 1 to 10 with higher scores indicating better function [22]. The scoring was rated by a trained examiner and rechecked by an assistant professor of psychiatry.

The CDT exhibits good psychometric properties, with sensitivity ranging from 86% to 97% and specificity ranging from 65% to 85% in neurocognitive assessments. Its construct validity is supported by its sensitivity to executive dysfunction and visuospatial deficits [22,23], and external validity is evidenced by correlations with comprehensive cognitive batteries like the Mini-Mental State Examination (MMSE) [23]. In SCZ, the CDT has been validated as a screening tool for cognitive impairment, though its specificity may be lower in psychiatric versus organic disorders [24].

### Statistical analysis

All analyses were performed using the R version 4.2.2 software. The categorical variables were described as frequency and percentage, and the continuous variables as mean (±SD). Normally distributed data were analyzed using paired T-test and independent T-test. Otherwise, a Wilcoxon or Mann-Whitney test was employed. In addition, multiple regression was conducted to assess predictive factors of changes in PANSS-positive symptoms, PANSS-negative symptoms, CDT score, and CGI score. A P-value < 0.05 was used to denote statistical significance.

### Results

A total of 84 patients participated in the study, of whom 60.7% were male. Forty-two (50%) patients were diagnosed with SCZ, and the remaining were BDP patients. Participants' mean (±SD) age was 42.85 (±11.51) years, ranging from 20 to 65 years old. Demographic characteristics of the participants are presented in Table 1 (Table 1). There were no statistical differences in gender distribution and age between SCZ and BDP patients. All the patients received occupational therapy (OT) during the hospitalization. However, they received different biological treatments (medication and ECT). You can have access to the dataset of the study via supporting information section (S1 Dataset).

At the time of admission, there were no significant differences between the SCZ and BDP patients in terms of the scores obtained in CDT, P-PANSS, and CGI assessments. However, SCZ patients had significantly higher baseline PANSS-negative scores compared to the BDP patients (p < 0.01).

At the time of discharge, no differences were observed in the CDT and PANSS-positive between the two groups. Nevertheless, SCZ patients scored higher in the PANSS-Negative (p < 0.01) and CGI (p < 0.01) scales at the time of discharge (Table 2).

**Table 1. Demographic characteristics of the participants.**

| Variable | | MeanSD/Median[min-max]/ No.[%] | | 95% CI | Effect size | p |
|---|---|---|---|---|---|---|
| | | Groups(n=84) | | | | |
| | | Schizophrenia | BDP | | | |
| Age(y) | | 39[24-65] | 40.7410.82 | [0.0035-0.25] | 0.02(small) | p=0.85 |
| Number of participants | | 42[%50] | 42[%50] | – | – | – |
| Gender | Female: 33[%39.3] | 15[%35] | 18[%42] | – | Cohen's w: 0.26(moderate) | p < 0.05 |
| | Male: 51[%60.7] | 27[%65] | 24[%58] | | | |
| ECT | | 10[%24] | 6[%14] | – | Cohen's w: 0.09 (small) | P=0.26 |
| Number of Medication | | 2[1-4] | 2[1-3] | [0.13-0.53] | 0.34(moderate) | p < 0.01 |
| Mood stabilizer | | 10[%24] | 38[%90] | – | Cohen's w: 0.64(large) | p=6.7e-10 |
| Antipsychotic medication | | 42[%100] | 40[%95] | – | – | – |

**Table 2. Comparison of Cognitive and Clinical Measures in Schizophrenia (SCZ) and Bipolar Disorder with psychotic features (BDP) Patients.**

| Assessment | SCZ Mean (SD) or Median [Range] | BDP Mean (SD) or Median [Range] | P-value | 95% CI | Effect Size |
|---|---|---|---|---|---|
| **Clock Drawing Test (CDT)** | | | | | |
| Admission CDT | 5 [2–10] | 5 [1–10] | p = 0.53 | [0.002–0.28] | 0.068 (small) |
| Discharge CDT | 9 [4–10] | 10 [4–10] | p = 0.07 | [0.02–0.39] | 0.19(small) |
| **P-value (Admission vs. Discharge)** | – | – | p < 0.05 (SZ) p < 0.001 (BDP) | [0.55–0.76] (SZ) [0.65–0.81] (BDP) | 0.66 (Large) (SZ) 0.74 (large) (BDP) |
| **PANSS-Positive** | | | | | |
| Admission PANSS-Positive | 35.76 (6.38) | 39 [10–46] | p = 0.14 | [0.01–0.36] | 0.16 (Small) |
| Discharge PANSS-Positive | 15.19 (4.12) | 14 [0–30] | p = 0.22 | [0.01–0.36] | 0.13 (Small) |
| **P-Value (Admission vs. Discharge)** | – | – | p < 0.0001 (SZ) p < 0.0001 (BDP) | [-4.39– -2.89] (SZ) [0.78–0.87] (BDP) | 3.48 (Large) (SZ) 0.87 (Large) (BDP) |
| **PANSS-Negative** | | | | | |
| Admission PANSS-Negative | 26.52 (9.36) | 20.98 (6.44) | p < 0.01 | [-1.15 – -0.28] | −0.69 (Large) |
| Discharge PANSS-Negative | 13 [7–33] | 9.56 [6–31] | p < 0.01 | [0.06–0.49] | 0.29 (Small) |
| **P-value (Admission vs. Discharge)** | – | – | p < 0.0001 (SZ) p < 0.0001 (BDP) | [0.87–0.87] (SZ) [0.85–0.87] (BDP) | 0.87 (Large) (SZ) 0.69 (Moderate) (BDP) |
| **Clinical Global Impression (CGI)** | | | | | |
| Admission CGI | 6 [1–7] | 6 [1–7] | p = 0.8 | [0.001–0.26] | 0.03 (Small) |
| Discharge CGI | 3 [2–6] | 2 [2–4] | p < 0.01 | [0.09–0.5] | 0.31 (Moderate) |
| **P-value (Admission vs. Discharge)** | – | – | p < 0.0001 (SZ) p < 0.0001 (BDP) | [0.88–0.92] (SZ) [0.89–0.93] (BDP) | 0.89 (Large) (SZ) 0.91 (Large) (BDP) |

\* P-value corresponds to between-group comparisons. # P-value corresponds to the admission vs. discharge comparison within each patient group. Data are presented as mean (SD) and median [range]

Comparison of the admission and discharge scores in each group CDT score increased significantly in both SCZ (p < 0.05) and BDP (p < 0.001) patients at the time of discharge compared to the time of admission. Both SCZ and BDP patients demonstrated significantly decreased scores in PANSS-positive (p < 0.0001), PANSS-negative (p < 0.0001), and CGI (p < 0.0001) assessments at the time of discharge when compared to the time of admission (Table 2) and Fig 1.

We further compared the mean changes in each test score (discharge vs. admission) between the two groups. Findings demonstrated that the CDT (p < 0.05), PANSS-positive (p < 0.05), and CGI (p < 0.01) scores were significantly higher in the BDP patients. However, the change in PANSS-negative (p < 0.05) was significantly higher in SCZ patients (Table 3).

A multiple linear regression analysis was conducted to examine the association between various clinical, treatment and demographic factors with changes in assessment scores.

For the predictive factor of changes in CDT score, the regression model explained approximately 51.1% of the variance in CDT improvement (R² = 0.5107, Adjusted R² = 0.428), and the overall regression model was statistically significant ($F_{(12, 71)} = 6.176$, p < 0.001). Among the predictors, the initial CDT score was significantly associated with changes in the CDT performance (β = −0.4497, p < 0.001), indicating that patients with lower initial CDT scores showed greater improvements post-hospitalization. The intercept was also statistically significant (β = 10.1003, p < 0.01), suggesting a baseline effect in CDT changes. However, other clinical variables, including the diagnosis of SCZ (β = −0.80174, p = 0.1063), PANSS scores, CGI scores, medication use, and demographic variables (age and gender), did not show statistically significant associations with CDT improvement.

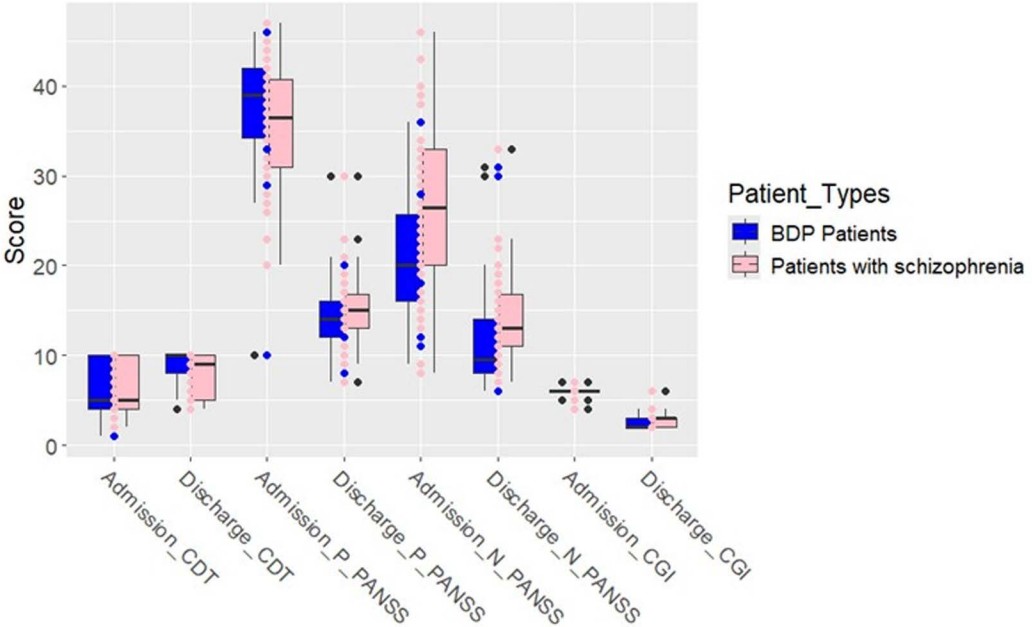

**Fig 1. Comparison of cognitive and clinical scores at admission and discharge between bipolar disorder with psychotic features (BDP) and schizophrenia patients.** Measures include the Clock Drawing Test (CDT), PANSS (Positive and Negative), and Clinical Global Impression (CGI) scores.

**Table 3. Comparison of clinical score changes from admission to discharge between schizophrenia (SCZ) and bipolar disorder with psychotic features (BDP) patients, including Clock Drawing Test (CDT), PANSS (Positive and Negative), and Clinical Global Impression (CGI) scores.**

| Assessment | SCZ | BDP | P-value | %95 CI | Effect size |
|---|---|---|---|---|---|
| CDT change | 0 [0-6] | 1.5 [0-7] | p<0.05 | [0.03– 0.43] | Mann-Whitney, two-sample rank-sum test(r): 0.22 (small) |
| PANSS-positive change | 20.57 (5.91) | 23 [3- 13] | p<0.05 | [0.05– 0.44] | Mann-Whitney, two-sample rank-sum test(r): 0.26 (small) |
| PANSS-negative change | 12.48 (7.09) | 9.09 (5.94) | p<0.05 | [-0.99–0.09] | Cohen's d: −0.51(moderate) |
| CGI change | 3 [0-5] | 4 [2-4] | p<0.01 | [0.05– 0.49] | Mann-Whitney, two-sample rank-sum test(r): 0.29(small) |

Data are presented as mean (SD), and median [Range].

For the predictive factor of changes in PANSS-positive symptoms, the regression explained 68.1% of the variance (Adjusted $R^2$=0.627, p<0.001), indicating a strong predictive model. The initial PANSS-positive score was the only significant predictor (β=0.72381, p<0.001), suggesting that patients with higher initial positive symptom severity had greater symptom reductions.

For the predictive factor of changes in PANSS-negative symptoms, the model explained 64.8% of variance (Adjusted $R^2$=0.589, p<0.001), suggesting a strong relationship between baseline negative symptoms and discharge improvement. The initial PANSS-negative score was a significant predictor (β=0.59116, p<0.001), showing that patients with more severe negative symptoms at admission tended to have greater reductions.

Since SCZ patients had significantly higher baseline PANSS-negative scores, we conducted an ANCOVA to compare post-treatment PANSS-negative scores between the two groups while controlling for baseline levels. The analysis revealed that even after adjustment, the groups differed significantly at discharge ($F_{(1, 81)}$ = 5.10, $p < 0.05$). Baseline PANSS-negative scores also had a strong effect on discharge outcomes ($F_{(1, 81)}$ = 42.61, $p < 0.001$). The adjusted

post-treatment means were similar between groups: 13.0 for both SCZ and BDP, with overlapping confidence intervals, indicating comparable symptom severity after treatment when baseline differences were accounted for.

For the predictive factor of changes in CGI, the model explained 41.7% of the variance (Adjusted $R^2 = 0.319$, $p < 0.001$), making it the weakest among the models. The initial CGI score was the only significant predictor ($\beta = 0.2595$, $p < 0.001$), indicating that patients with more severe initial clinical impressions showed greater improvement.

## Discussion

Overall, our findings demonstrated that in SCZ and BDP patients, the cognitive function, severity of positive and negative symptoms, and clinical global impression improved following treatment. The extent of improvements observed in positive symptoms and global impression was higher in BDP patients, while SCZ patients demonstrated higher changes in negative symptoms. Notably, we demonstrated that BDP patients had significantly more pronounced changes in CDT scores compared to the SCZ patients and the CDT scores at discharge were relatively higher than SCZ patients. Interestingly, we showed that baseline symptom severity plays a crucial role in determining clinical and cognitive improvements post-hospitalization.

Generally, CDT is a sensitive test capable of identifying the extensive disruption of brain systems that is characteristic of numerous neurocognitive disorders. This test is frequently employed to diagnose dementia [25]. Recently, the application of CDT has broadened to include the detection of cognitive impairments in other neurological conditions, such as hypertension-induced brain damage and focal brain damage in individuals suffering from traumatic brain injuries and strokes [15,26,27]. Additionally, it has been utilized to assess cognitive deficits in SCZ [12,28]. Despite the initial similarities in CDT scores between SCZ and BDP patients at the start of treatment, our findings demonstrated more pronounced improvements in the cognitive function of the BDP group at the end of treatment, exceeding those seen in the SCZ group. The CDT scores were even marginally significantly higher in the BDP group at the time of discharge. These findings partly align with prior research indicating that cognitive impairments are more severe in SCZ patients compared to those suffering from BD [29]. Moreover, the lower extent of cognitive improvements in SCZ patients compared to BDP in our study confirms these observations. It's important to highlight that the deficit patterns vary significantly among different psychiatric conditions. In SCZ, cognitive deficits are more pronounced than in BDP, and these deficiencies manifest even before symptoms appear, a phenomenon not observed in BD [10,30,31].

The differential cognitive and clinical improvements observed may also be influenced by the biological treatments and interventions provided during hospitalization, which varied between groups. All participants received OT twice weekly (45-minute sessions), aimed at improving functional skills, while biological treatments differed: SCZ patients were primarily treated with antipsychotics, and BDP patients received a combination of mood stabilizers (e.g., lithium, valproate) and antipsychotics, with some of both groups receiving ECT for severe symptoms. Literature suggests that second-generation antipsychotics, commonly used in SCZ, can yield modest cognitive improvements, but these gains are often limited by the disorder's underlying neurobiology [32,33]. In contrast, mood stabilizers and antipsychotics in BDP may facilitate greater cognitive recovery, particularly during manic episodes, by stabilizing mood and reducing psychotic symptoms, with studies reporting moderate effect sizes (≈ 0.4–0.5) on executive function and processing speed post-treatment. ECT, used in some BDP patients, has been associated with rapid symptom reduction but can cause transient cognitive side effects, though these typically resolve within weeks [34]. OT, which focuses on daily living skills and social engagement, may have contributed to improvements in negative symptoms by enhancing functional capacity, with prior research indicating small to moderate effects on social cognition in both disorders [35]. However, the uniform application of OT and the lack of detailed medication data (e.g., dosages) in our study limit our ability to fully disentangle their contributions to the observed outcomes, particularly the greater CDT gains in BDP. Future studies should examine the specific effects of treatment type and intensity on cognitive and clinical trajectories in these populations.

Generally, both SCZ and BDP can manifest both positive and negative symptoms of psychosis. However, as per a comprehensive review, the lifetime and current incidence rates of various psychotic symptoms in BD indicate that these symptoms are less prevalent than those observed in SCZ [36]; despite these findings, our analysis indicates that individuals with SCZ exhibit a higher severity of negative symptoms compared to those diagnosed with BDP. As mania is characterized by increased energy, verbosity, and goal-oriented behavior, this difference becomes more apparent [37]. However, the severity of positive psychotic symptoms was found to be similar across both groups.

Despite ongoing research, the connection between cognitive deficits and psychotic symptoms in individuals suffering from SCZ and BDP remains elusive. Our study reveals that cognitive deficits can serve as a distinct factor separated from psychotic characteristics in both conditions, as we found no correlations between the changes in CDT score and the changes in PANSS scores and the initial cognitive state of the patients' initial CDT score) were the best predictors of the final cognitive outcome. Positive psychotic symptoms like hallucinations and delusions are generally believed to be linked to cognitive impairments, including difficulty focusing, recalling information, and organizing tasks [36]. Delusions have also been found to be associated with altered activity of certain cortical and subcortical brain regions [38]. Negative symptoms such as paranoia, suspicion, and fear can also cause cognitive deficits, leading to constant anxiety and tension [39]. On the other hand, it has been suggested that the extent of SCZ's cognitive deficits is complex and not solely dependent on positive or negative psychotic symptoms. They are influenced by factors like genetic predisposition, environment, and physical health [10]. Cognitive impairment is also reported in BD, evident even before the onset of the disease and throughout its acute and remitted stages [31]. Some reports suggest that processing speed is the domain most affected in SCZ and that processing speed deficits are the strongest predictor of general cognitive performance [10]. Other studies have highlighted deficits in attention, memory, reasoning, and executive functioning [40].

Although we did not directly assess CDT's ability to differentiate organic and non-organic (psychiatric) brain disorders, as a secondary finding of our study, it is plausible that CDT might not be an ideal test to differentiate the two disease entities. In our previous study using the same CDT scoring, we showed that the mean (SD) score of CDT was 7.52 (2.82) in a group of patients with brain lesions due to stroke, traumatic brain injury, brain tumor, or brain aneurysm surgery [15], which aligns with decreased scores seen in SCZ and BDP in the current study. This overlap in CDT performance between psychiatric patients and those with structural brain lesions suggests that likely the CDT cannot reliably distinguish the cognitive deficits arising from these two different pathological substrates.

Our study also demonstrated the association between baseline symptom severity and clinical and cognitive improvements. Accordingly, initial symptom severity was the strongest predictor of improvement for both positive and negative PANSS symptoms, CGI, and cognitive performance (in terms of CDT). The PANSS symptoms (especially for positive symptoms) showed the highest predictive strength, while CGI had the weakest model fit. These findings suggest that baseline symptom severity plays a crucial role in determining clinical and cognitive improvements post-hospitalization. These findings are consistent with previous research, which has shown that baseline symptom severity is a key determinant of treatment response and clinical outcomes in psychiatric disorders [41–43].

Overall, our research did not uncover any significant relationships between changes in CDT scores and changes in other scales, suggesting that cognitive deficits could act as findings irrespective of positive and negative symptoms. We recommend that clinicians integrate brief, validated cognitive screening tools, such as the CDT used in our study or the Montreal Cognitive Assessment (MoCA), into standard assessments at admission, during hospitalization, and at follow-up visits (e.g., every 3–6 months). These tools can help identify cognitive deficits early, track changes over time, and inform treatment planning. For treatment, our study highlights the potential of cognitive interventions, which we briefly mentioned earlier in the manuscript as an effective approach for these disorders. Specifically, cognitive remediation therapy (CRT), a structured intervention involving exercises to improve attention, memory, and executive functioning, has shown promise in SCZ and BDP [44]. We suggest that clinicians incorporate CRT into inpatient and outpatient management protocols alongside OT.

This study has several limitations that should be considered when interpreting the results. First, the sample size was relatively small (n = 42 per group), which may limit the generalizability of the findings. A larger sample could better detect subtle cognitive differences between groups. Second, this was a cross-sectional study assessing patients only at admission and discharge. A longitudinal design with extended follow-up could better characterize long-term cognitive and clinical trajectories. Third, cognitive assessment relied solely on the CDT, which, while sensitive, may not capture the full spectrum of cognitive domains; additional neurocognitive tests (e.g., assessing memory, and attention) could provide a more comprehensive evaluation. Fourth, the lack of healthy controls prevented us from determining whether cognitive performance normalized with treatment. Fifth, we did not systematically record hospitalization duration for all participants, retrieving data for only a subset of 25 patients (mean = 21.28 days), which precluded analysis of its impact on outcomes; longer stays could theoretically enhance treatment exposure and influence recovery. Sixth, the study was conducted during the COVID-19 pandemic (September 2021 to December 2022), and while we included only COVID-19-negative patients, the broader context (e.g., delayed care, hospital constraints) may have influenced symptom severity at admission (e.g., high baseline PANSS scores). Seventh, the lack of detailed medication data (dosages, adherence) limited our ability to assess treatment effects on outcomes, particularly given the unreliability of pre-admission self-reports in this population. Eighth,we did not explore potential gender differences in presentation or outcomes, though we hypothesize that men may present with poorer outcomes due to social or cultural factors affecting treatment-seeking behavior in Iran; this warrants further investigation in future studies.Larger, multi-center studies with longitudinal follow-up, comprehensive neurocognitive batteries, detailed treatment data, and control groups are warranted to confirm and extend these findings. Despite these limitations, this study provides preliminary evidence of distinct cognitive and clinical recovery trajectories in SCZ and BDP, emphasizing the importance of regular cognitive assessment in their management.

## Conclusions

Taken together, our findings suggest that patients with SCZ showed relatively less improvement in the CDT compared to those with BDP following hospitalization. This may indicate differences in cognitive recovery trajectories between the two groups rather than an inherent cognitive deficit in SCZ. Additionally, cognitive improvements were not correlated with changes in positive or negative symptoms in either group but appeared to be influenced by the initial cognitive profile. Given that the CDT is primarily a screening tool rather than a comprehensive cognitive assessment, these results should be interpreted with caution. Future research incorporating more extensive neurocognitive testing is needed to better characterize cognitive trajectories in SCZ and BDP.

## Supporting information

**S1 Dataset.** **This is the dataset of the study.**
(XLSX)

## Author contributions

**Conceptualization:** Ahmad Pirani, Atiye Sarabi-Jamab, Faeze Bostan, Razieh Salehian, Mehran Ilaghi, Ali Aledavoud, Seyed Vahid Shariat, Fatemeh sadat Mirfazeli.

**Data curation:** Ahmad Pirani, Atiye Sarabi-Jamab, Faeze Bostan, Razieh Salehian, Fatemeh sadat Mirfazeli.

**Formal analysis:** Atiye Sarabi-Jamab.

**Investigation:** Ahmad Pirani, Atiye Sarabi-Jamab, Seyed Vahid Shariat.

**Methodology:** Faeze Bostan, Razieh Salehian.

**Supervision:** Fatemeh sadat Mirfazeli.

**Validation:** Razieh Salehian, Seyed Vahid Shariat, Fatemeh sadat Mirfazeli.

**Writing – original draft:** Mehran Ilaghi, Ali Aledavoud.

**Writing – review & editing:** Razieh Salehian, Seyed Vahid Shariat, Fatemeh sadat Mirfazeli.

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
