## [Decision Letter · Decision Letter 0]

PONE-D-24-39274Differential Cognitive and Clinical Improvements in Schizophrenia and Bipolar Disorder Following Hospitalization: A Comparative Analysis Based on the Clock Drawing TestPLOS ONE

Dear Dr. Mirfazeli,

Thank you for submitting your manuscript to PLOS ONE. After careful consideration, we feel that it has merit but does not fully meet PLOS ONE’s publication criteria as it currently stands. Therefore, we invite you to submit a revised version of the manuscript that addresses the points raised during the review process.

This holds an interesting potential to the field and I read with enthusiasm. Nevertheless, there are several concerns from definitions to proper interpretation of the findings. I will list as follows:

1) Please double-check grammar (e.g. punctuation and verb tense);

2) Please double-check refs (e.g. formatting and punctuation);

3) The use of convenience sampling without justification introduces selection bias and limits generasalitions. Also, the absence of controls further weakens the findings. It'd be interesting a more representative sampling method or refining the existing approaches. For instance, the sample size is vague, relying on an effect size from a prior study without clear assumptions or a power analysis. The explanation of its selection, the effects selection and the CIs are essential. The lack of details on inter-rater reliability introduces risks to data accuracy. Overall, both Methods and Results need to be considerably reshaped to ensure a rigorous and transparent study. Also, consider that 1. There is no mention of controlling for medication, comorbidities, or treatment duration, which influence the observed differences, 2. Values are provided without effect sizes or CIs, making the results harder to interpret, 3. There is an inconsistency in how SDs are reported as well as decimals, 4. The tables lack clear labelling, making it difficult to interpret results, and there aren't graphs to visually represent the data. Please consider refining those too;

4) Results:

- Admission

The intriguing difference between symptom burden in patients aligns with clinical understanding. The interesting thing is the less proeminence within a group with several differences in meds, especially poly use of meds. These comparable outcomes in test kind indicate the notion that there are impairments regardless of the condition, albeit through potentially different mechanisms. However, although important, both tests won't provide a bigger picture to subtle differences in functional outcomes. Comparable global severity and cognitive impairments in both groups highlight the importance of addressing functional outcomes across groups. To strengthen this section, the authors could consider including merged symptom indices, interaction, any possible influence of PANSS, pos and negative, across the tests, etc;

- Discharge

- Trends in PANSS negative symptoms and scores hint subtle differences that might warrant investigation with larger sample sizes. As mentioned, understanding the correlations and interactions of the discharge measurements would be really interesting to have a comprehension of the gap or what could have occurred to explain the differences. Examining predictors, correlating changes, and subgroup analyses provide more detailed insights into the findings;

- The paired tests should be better explored to explain the main clinical implications of the data. Link specific meds to outcomes (e.g. PANSS scores, CGI changes) to evaluate the efficacy of different treatments in each group - even subtle;

- Overall, PANSS differences at admission predict scores at discharge, with negative symptoms showing stronger consistency. This is important, although completely different from changes in CGI, for example, where the pos symptoms were contundent;

5) I calculated and noticed that differences were closely linked to baseline severity. Higher baseline positive symptoms strongly predicted changes in positive symptoms (R² = 0.639, p < 0.001), while cognitive had minimal influence. This was interesting in CGI when baseline negative symptoms were higher (interaction p = 0.019). Patients starting with more severe positive symptoms had improvements, which is counterintuitive. For instance, for delta PANSS the differences were really worth investigating. The meds also influenced the outcomes. While running PCA,  I observed that 91.6% of the variance in PANSS items was explained by three components, with PC1 (48.3%) likely reflecting overall symptom severity. PC1 correlated with changes in positive symptoms and global severity. This is aligning with observed data a few years ago, but the lack of logical explanation between PANSS and other variables with the fluctuation in scores are worrying. Please expand and improve the Results and Discussion;

6) Overall, without proper refinement it's difficult to assess the relevance of the study;

Thank you for your valuable submission.

Wishing you success with the study.

We look forward to receiving your revised manuscript.

Kind regards,

Thiago P. Fernandes, PhD

Academic Editor

PLOS ONE

Journal requirements:When submitting your revision, we need you to address these additional requirements.

1. Please ensure that your manuscript meets PLOS ONE's style requirements, including those for file naming. The PLOS ONE style templates can be found at https://journals.plos.org/plosone/s/file?id=wjVg/PLOSOne_formatting_sample_main_body.pdf and https://journals.plos.org/plosone/s/file?id=ba62/PLOSOne_formatting_sample_title_authors_affiliations.pdf.

2. Please describe in your methods section how capacity to provide consent was determined for the participants in this study. Please also state whether your ethics committee or IRB approved this consent procedure. If you did not assess capacity to consent please briefly outline why this was not necessary in this case.

3. We note that there is identifying data in the Supporting Information file <Data.xlsx>. Due to the inclusion of these potentially identifying data, we have removed this file from your file inventory. Prior to sharing human research participant data, authors should consult with an ethics committee to ensure data are shared in accordance with participant consent and all applicable local laws.

-Location data

Please remove or anonymize all personal information (<name and age>), ensure that the data shared are in accordance with participant consent, and re-upload a fully anonymized data set. Please note that spreadsheet columns with personal information must be removed and not hidden as all hidden columns will appear in the published file.

Reviewers' comments:

Reviewer's Responses to Questions

**Comments to the Author**

1. Is the manuscript technically sound, and do the data support the conclusions?

Reviewer #1: Partly

Reviewer #2: Yes

2. Has the statistical analysis been performed appropriately and rigorously? 

Reviewer #1: No

Reviewer #2: Yes

3. Have the authors made all data underlying the findings in their manuscript fully available?

Reviewer #1: Yes

Reviewer #2: Yes

4. Is the manuscript presented in an intelligible fashion and written in standard English?

Reviewer #1: Yes

Reviewer #2: Yes

5. Review Comments to the Author

Reviewer #1: The study, "Differential Cognitive and Clinical Improvements in Schizophrenia and Bipolar Disorder Following Hospitalization: A Comparative Analysis Based on the Clock Drawing Test" by Ahmad Pirani et al., examined changes in cognitive function among hospitalized patients with Schizophrenia and Bipolar Disorder with Psychotic Features (BDP) using the Clock Drawing Test (CDT). This study included 84 patients, equally divided between those with Schizophrenia and those with BDP. Patients were evaluated at hospital admission and discharge using the Positive and Negative Syndrome Scale (PANSS) and the Clinical Global Impression (CGI). The results indicated that patients with Schizophrenia showed very little improvement in cognitive function after treatment compared to those with BDP, suggesting a potential inherent cognitive deficit associated with Schizophrenia. However, the study lacks information about the treatment or medications provided, and the small sample size limits the strength of its conclusions. Moreover, the study design was not sufficiently robust to fully address the stated hypothesis. The authors also did not assess other critical cognitive domains, such as attention, memory, reasoning, and executive functions. Overall, while the findings are suggestive, additional research with more comprehensive methodologies and larger samples is needed.

Reviewer #2: Dear Editor,

I appreciate the opportunity to review the manuscript entitled:

"Differential Cognitive and Clinical Improvements in Schizophrenia and Bipolar Disorder Following Hospitalization: A Comparative Analysis Based on the Clock Drawing Test"

I commend the authors for describing this critical and timely issue. The paper is interesting and well-written; however, I would like to highlight some issues that merit revision:

In the manuscript, it is not detectable whether an assessment was made of what might be a protective factor through psychotherapy and counseling intervention, which people often use without prescription to prevent or alleviate symptoms at the onset. I would ask the authors if this was evaluated during the survey and add a short paragraph on this issue; if data are unavailable, add them to the limitations.

6. PLOS authors have the option to publish the peer review history of their article (what does this mean? ). If published, this will include your full peer review and any attached files.

**Do you want your identity to be public for this peer review?** For information about this choice, including consent withdrawal, please see our Privacy Policy .

Reviewer #1: No

Reviewer #2: No

---

## [Author Response · Author response to Decision Letter 1]

4 Mar 2025

We have attached a response to reviewer file with detailed response to the comments of the editor and two reviewers.

---

## [Decision Letter · Decision Letter 1]

PONE-D-24-39274R1Differential Cognitive and Clinical Improvements in Schizophrenia and Bipolar Disorder Following Hospitalization: A Comparative Analysis Based on the Clock Drawing TestPLOS ONE

Dear Dr. Mirfazeli,

Thank you for submitting your manuscript to PLOS ONE. After careful consideration, we feel that it has merit but does not fully meet PLOS ONE’s publication criteria as it currently stands. Therefore, we invite you to submit a revised version of the manuscript that addresses the points raised during the review process.

Thank you for your valuable submission.

This is indeed a good and interesting study.

I encourage the authors to carefully review all comments, and ensure that the rebuttal clearly outlines both the changes made and any aspects that remain unchanged, with detailed justifications. Please bear in mind that our aim is to avoid an overly lengthy process, and we cannot guarantee further rounds if the revisions do not adequately address the reviewers’ concerns.

If the authors are willing to proceed, I would be happy to re-read and continue with.

We look forward to receiving your revised manuscript.

Kind regards,

Thiago P. Fernandes, PhD

Academic Editor

PLOS ONE

Reviewers' comments:

Reviewer's Responses to Questions

**Comments to the Author**

1. If the authors have adequately addressed your comments raised in a previous round of review and you feel that this manuscript is now acceptable for publication, you may indicate that here to bypass the “Comments to the Author” section, enter your conflict of interest statement in the “Confidential to Editor” section, and submit your "Accept" recommendation.

Reviewer #1: All comments have been addressed

Reviewer #2: All comments have been addressed

Reviewer #3: (No Response)

2. Is the manuscript technically sound, and do the data support the conclusions?

Reviewer #1: Yes

Reviewer #2: Yes

Reviewer #3: Partly

3. Has the statistical analysis been performed appropriately and rigorously? 

Reviewer #1: Yes

Reviewer #2: Yes

Reviewer #3: No

4. Have the authors made all data underlying the findings in their manuscript fully available?

Reviewer #1: Yes

Reviewer #2: Yes

Reviewer #3: Yes

5. Is the manuscript presented in an intelligible fashion and written in standard English?

Reviewer #1: Yes

Reviewer #2: Yes

Reviewer #3: Yes

6. Review Comments to the Author

Reviewer #1: (No Response)

Reviewer #2: The paper is very interesting and well-written, methodologically unexceptionable, and the new implementations provide a valid contribution to the work. Every requested correction has been done, and the manuscript is now suitable for publication.

Reviewer #3: It is also uploaded and attached.

Thank you for the opportunity to review the manuscript, "Differential Cognitive and Clinical Improvements in Schizophrenia and Bipolar Disorder Following Hospitalization: A Comparative Analysis Based on the Clock Drawing Test," for PLOS One.

The purpose of this study was to examine changes in cognitive function and associated positive and negative symptoms among participants with schizophrenia and bipolar disorder with psychotic features (BDP). This study was cross-sectional and included 84 participants who were hospitalized at two psychiatric hospitals; participants were equally divided between both groups. Participants were evaluated at two timepoints (hospital admission and discharge) using the Positive and Negative Syndrome Scale (PANSS), the Clinical Global Impression (CGI), and the Clock Drawing Test (CDT). Within- and between- group changes on the outcome variables were assessed. The results indicated that participants with schizophrenia evidenced less improvement in cognitive function at post-treatment compared to participants with BDP, regardless of change in positive and negative symptoms but related to baseline cognitive profile. Findings suggest underlying cognitive deficits associated with schizophrenia.

I appreciate the focus of this study—as it appears that research examining cognitive functioning in bipolar disorder is limited. The manuscript is written clearly and the preliminary findings from this study have the potential for scientific contributions, which may later inform future research and clinical care. The manuscript may be further strengthened by the following comments and suggestions. As such, I am recommending that the following areas be addressed before the manuscript is considered for publication in PLOS One.

General

1. Please check that the manuscript is in accordance to APA style 7th Edition (e.g., spell out all acronyms, report p values less than .001 as “p < .001,” etc.)

2. Please be mindful of consistency in formatting (e.g., the first letters of “clock drawing test” are capitalized in some places in lowercase in other places, etc.)

Abstract

1. It might be helpful for the abstract to include information about participant demographics (e.g., age, race/ethnicity, gender) and to note the statistical methods employed.

2. Consider tempering language/conclusions since the study is examining associations among outcome variables. This is particularly important because the Clock Drawing Test is typically used as a screener for cognitive impairment in participants with schizophrenia. From this one score, I would use caution in concluding that less pronounced improvements on one cognitive screening measure suggest underlying cognitive deficits for participants with schizophrenia.

Introduction

1. Overall, the introduction is easy to read and well written. I appreciate the general literature review and that the authors acknowledged gaps in the literature and provided rationale for how the study addresses those gaps.

2. One of the aims of this study is to measure cognitive and clinical improvements after treatment. Although briefly addressed in the Discussion, the Introduction does not review relevant treatments and expected treatment response/clinical outcomes for these disorders, especially with regard to the study’s primary outcomes (i.e., positive/negative symptoms, cognition). Inclusion of this background information would provide a better context for the study results and would facilitate a discussion of potential clinical applications of study results in the Discussion.

3. Please clearly state study hypotheses and predictions that accompany study objectives.

Method

1. The timeframe that these data were collected is September 2021 to December 2022. Given that some data were collected during COVID-19, I’m wondering how the historical context may or may not affect study results (e.g., potentially exacerbate disease symptoms).

2. The credentials of the people making the schizophrenia and BDP diagnoses and how the diagnoses were made (e.g., comprehensive evaluation? diagnostic clinical interviewing? by history?) to determine if a participant met diagnostic criteria are not clear.

3. Participants were assessed at two timepoints (admission and discharge); however, it is not clear how many days participants were hospitalized during that timeframe. It would be helpful to include the average time (and SD) from admission to discharge for each group. I’m wondering if duration of hospital stay could be a factor influencing study results.

4. Authors state that all participants received occupational therapy. How often/how much OT did participants receive? I’m wondering if OT intervention duration plays a role in disorder symptoms and cognitive outcomes.

5. It was reported that participants received different biological treatments (i.e., medication and ECT). However, it is unclear how many participants received medication, ECT, or both treatments. What proportion of participants received medication, ECT, or a combination that could alter the frequency/severity of their symptoms and level of cognitive functioning?

6. Of the participants taking medication, how many were on medication for pre- and post-testing? Perhaps medication type positively or negatively affects CDT test performance and scores on the other measures.

7. The authors report demographic data for participant age and gender, but no additional demographic data is provided (i.e., race/ethnicity, primary language, comorbidity, SES, etc.). It is unclear if groups differ on these additional pre-treatment variables; not providing these data notably limit interpretation and generalizability of study results.

8. Relatedly, the demographic data for the total sample is provided in Table 1 but it is not provided for each group. Table 1 could be enhanced by reporting demographic data for both samples. Please see an example below:

Variable Schizophrenia

(n=42) BDP

(n=42) 95% CI Effect size p

M SD M SD

Gender (male/female) 51/33 / --

Age

Race/ethnicity (W/B/H/M) --

SES

Biological treatment status

Medication (No/Yes) --

ECT (No/Yes) --

Combination (No/Yes) --

Duration of OT (minutes)

Comorbidity (No/Yes) --

Duration of hospital stay (days)

9. I appreciate the detailed information about the instruments used in this study. Please also include the range of possible scores for each measure as well as instrument psychometrics (i.e., evidence for reliability, internal and external validity).

10. Were there any outliers? If so, how were those defined and handled?

11. The authors reported comparing the mean changes in each test score (discharge vs. admission) between the two groups. I'm curious if the authors considered using pre-treatment scores as a covariate, especially since schizophrenia participants had significantly higher baseline PANSS-negative scores, to test whether post-treatment means, adjusted for pre-treatment scores, differed between the two groups.

12. This may be beyond the scope of the study, but I'm curious as to what a mixed-model ANOVA (group x symptom domain [positive, negative, cognitive, CDT] x time) would yield with regard to the interaction terms.

Results

1. Table 2 does not clearly show the data in an organized manner, and it is difficult to follow. Consider the following table:

Variable Schizophrenia

(n=42) BDP

(n=42) 95% CI Effect size p

M SD M SD

Pre-treatment/Admission

CDT

PANNS Positive

PANNS Negative

Post-treatment/Discharge

CDT

PANNS Positive

PANNS Negative

2. The severity of positive psychotic symptoms was found to be similar across both groups. Could this be a result of sampling bias (positive symptoms are likely more noticeable/impairing than negative symptoms, prompting hospitalization at a psychiatric facility)?

Discussion

1. I appreciate that the authors provide rationale and evidence from literature to explain why there were more pronounced improvements in cognitive function at post-treatment for the BDP group than for the schizophrenia group. However, the authors do not address the potential effects of biological treatment and intervention (i.e., OT) on these outcomes. Perhaps the authors could comment on treatment/intervention factors that might influence primary outcomes, using previous literature on expected treatment response/clinical outcomes for these disorders in the explanation.

2. I appreciate that the authors provide an implication for clinical practice. However, the implication seems relatively general and to lack clear, detailed guidelines for actionable steps, which limits the applicability of findings for researchers and practitioners. It would be helpful to expand this implication and to provide guidance for researchers and practitioners to follow. For example, earlier in the manuscript the authors mention cognitive intervention as a potential effective treatment for these disorders, but it was not further addressed in the Discussion as a potential future direction for the field.

3. In terms of the limitations, I appreciate that the authors acknowledge that the study’s relatively small sample size is a limitation. However, the authors do not address that the majority of study participants identify as male, which also limits generalizability. I find myself looking to the context to help explain why the sample mostly identifies as male, but little context is provided.

4. Perhaps authors could comment on potential sampling bias and limits to generalizability with regard to collecting data during COVID-19 and to collecting data from study participants who were hospitalized. Another potential limitation may be that the study lacked a control group.

7. PLOS authors have the option to publish the peer review history of their article (what does this mean? ). If published, this will include your full peer review and any attached files.

**Do you want your identity to be public for this peer review?** For information about this choice, including consent withdrawal, please see our Privacy Policy .

Reviewer #1: No

Reviewer #2: No

Reviewer #3: No

---

## [Author Response · Author response to Decision Letter 2]

11 Apr 2025

We sincerely thank you and the reviewers for your valuable time and insightful comments on our manuscript. We appreciate the opportunity to revise and improve our work based on the constructive feedback we have received. We have carefully considered all the reviewers’ comments and have made the necessary revisions to the manuscript to address their suggestions. For clarity, we have provided a detailed point-by-point response to each comment below. Reviewer comments are presented in bold, followed by our responses in regular text. All changes made to the manuscript have been highlighted, and the revised phrases are also provided below for reference. We hope that the revised manuscript meets your expectations, and we are happy to provide further clarifications or revisions as needed.

Thank you for considering our revised manuscript for publication in PLOS ONE.

Comments to the Author

6. Review Comments to the Author

Reviewer #1: (No Response)

Reviewer #2: The paper is very interesting and well-written, methodologically unexceptionable, and the new implementations provide a valid contribution to the work. Every requested correction has been done, and the manuscript is now suitable for publication.

Reviewer #3: It is also uploaded and attached.

Thank you for the opportunity to review the manuscript, "Differential Cognitive and Clinical Improvements in Schizophrenia and Bipolar Disorder Following Hospitalization: A Comparative Analysis Based on the Clock Drawing Test," for PLOS One.

The purpose of this study was to examine changes in cognitive function and associated positive and negative symptoms among participants with schizophrenia and bipolar disorder with psychotic features (BDP). This study was cross-sectional and included 84 participants who were hospitalized at two psychiatric hospitals; participants were equally divided between both groups. Participants were evaluated at two timepoints (hospital admission and discharge) using the Positive and Negative Syndrome Scale (PANSS), the Clinical Global Impression (CGI), and the Clock Drawing Test (CDT). Within- and between- group changes on the outcome variables were assessed. The results indicated that participants with schizophrenia evidenced less improvement in cognitive function at post-treatment compared to participants with BDP, regardless of change in positive and negative symptoms but related to baseline cognitive profile. Findings suggest underlying cognitive deficits associated with schizophrenia.

I appreciate the focus of this study—as it appears that research examining cognitive functioning in bipolar disorder is limited. The manuscript is written clearly and the preliminary findings from this study have the potential for scientific contributions, which may later inform future research and clinical care. The manuscript may be further strengthened by the following comments and suggestions. As such, I am recommending that the following areas be addressed before the manuscript is considered for publication in PLOS One.

General

1. Please check that the manuscript is in accordance to APA style 7th Edition (e.g., spell out all acronyms, report p values less than .001 as “p < .001,” etc.)

Thank the reviewer for the precious feedback, we did that. We organized manuscript based on APA style 7th Edition and check other requests based on your useful comment.

2. Please be mindful of consistency in formatting (e.g., the first letters of “clock drawing test” are capitalized in some places in lowercase in other places, etc.)

Thank the reviewer for your precious feedback, we corrected all.

Abstract

1. It might be helpful for the abstract to include information about participant demographics (e.g., age, race/ethnicity, gender) and to note the statistical methods employed.

Thank the reviewer, we changed that as well:

“This cross-sectional study enrolled 84 patients (42 schizophrenia, 42 BDP, 51 male and 33 female) aged 42.85 (±11.51), ranging from 20 to 65 years old, from two psychiatric hospitals in Iran”

“Within-group and between-group changes were analyzed using paired t-tests for pre-post comparisons and multiple regression was used to assess predictive factors of changes in cognitive and symptom changes.”

2. Consider tempering language/conclusions since the study is examining associations among outcome variables. This is particularly important because the Clock Drawing Test is typically used as a screener for cognitive impairment in participants with schizophrenia. From this one score, I would use caution in concluding that less pronounced improvements on one cognitive screening measure suggest underlying cognitive deficits for participants with schizophrenia.

Thank the reviewer for pointing out this important matter.

“While both groups exhibited cognitive and clinical improvements following hospitalization, patients with schizophrenia showed relatively less improvement in the CDT compared to those with BDP. These findings may suggest that cognitive recovery may follow a different trajectory in schizophrenia, independent of changes in positive and negative symptoms but related to the initial cognitive profile. However, given that the CDT is a screening tool rather than a comprehensive cognitive assessment, these results should be interpreted with caution. Future research should incorporate broader neurocognitive assessments to better understand the cognitive trajectories of these populations “

Introduction

1. Overall, the introduction is easy to read and well written. I appreciate the general literature review and that the authors acknowledged gaps in the literature and provided rationale for how the study addresses those gaps.

We thank the reviewer for this comment

2. One of the aims of this study is to measure cognitive and clinical improvements after treatment. Although briefly addressed in the Discussion, the Introduction does not review relevant treatments and expected treatment response/clinical outcomes for these disorders, especially with regard to the study’s primary outcomes (i.e., positive/negative symptoms, cognition). Inclusion of this background information would provide a better context for the study results and would facilitate a discussion of potential clinical applications of study results in the Discussion.

Thank the reviewer for the valuable comments to make our paper better.

„Treatment approaches for SCZ and BDP typically involve a combination of pharmacotherapy, psychotherapy, and, in some cases, somatic interventions like electroconvulsive therapy (ECT), aimed at reducing symptom severity and improving functional outcomes. For SCZ, antipsychotic medications are the cornerstone of treatment, effectively reducing positive symptoms such as hallucinations and delusions within weeks, though negative symptoms and cognitive deficits often show limited improvement (13). A meta-analysis by Leucht et al. (2017) found that antipsychotics yield moderate effect sizes (Cohen’s d ≈ 0.5) for positive symptom reduction, but cognitive gains are less consistent, with some studies reporting minor improvements in processing speed and attention following second-generation antipsychotics (14). In BDP, mood stabilizers (e.g., lithium, valproate) combined with antipsychotics during psychotic manic episodes are standard, with evidence suggesting robust reductions in positive symptoms and mania-related cognitive disruptions within 4-6 weeks. A review by Nierenberg et al. (2023) highlights that bipolar patients often experience partial cognitive recovery post-treatment, particularly in executive functioning, though deficits may persist during euthymic phases (5). These differential treatment responses underscore the need to evaluate how clinical and cognitive outcomes align in hospitalized patients, providing a foundation for understanding potential therapeutic impacts in this study’s context.“

3. Please clearly state study hypotheses and predictions that accompany study objectives.

Done:

“We hypothesized that: (1) both schizophrenia and BDP patients would exhibit significant improvements in cognitive function (measured by CDT) and clinical symptoms (measured by PANSS and CGI) following hospitalization, given the expected efficacy of standard treatments; (2) BDP patients would demonstrate greater improvements in CDT scores and positive symptom reduction compared to schizophrenia patients, reflecting evidence of less severe baseline cognitive deficits and stronger treatment response in bipolar disorder; and (3) changes in cognitive function would not strongly correlate with changes in positive or negative symptoms in either group, suggesting that cognitive deficits may represent a distinct domain from psychotic symptomatology. “

These predictions were based on prior findings of differential cognitive impairment and treatment responsiveness between the two disorders (5, 10, 23)

Method

1. The timeframe that these data were collected is September 2021 to December 2022. Given that some data were collected during COVID-19, I’m wondering how the historical context may or may not affect study results (e.g., potentially exacerbate disease symptoms).

We thank the reviewer for highlighting the important historical context of the COVID-19 pandemic during our data collection period (September 2021 to December 2022) and its potential influence on study outcomes. To mitigate the direct impact of COVID-19 on our sample, we included only patients who tested negative for COVID-19 at admission, as confirmed by hospital protocols at Rasoul Akram Hospital and Iran Psychiatry Hospital. This exclusion criterion ensured that our findings were not confounded by the acute effects of the virus on symptom presentation or cognitive function in schizophrenia and BDP patients.

However, we acknowledge that the broader context of the pandemic may have indirectly influenced our results. During this period, public health measures, fear of infection, and strained healthcare systems likely affected patients’ access to routine psychiatric care, potentially leading to delayed treatment-seeking and more severe symptom presentations at admission. This aligns with literature suggesting that the COVID-19 pandemic exacerbated mental health outcomes, with studies reporting increased relapse rates and symptom severity in schizophrenia and bipolar disorder due to reduced outpatient care and heightened psychosocial stress . In our study, we observed relatively high baseline PANSS scores (e.g., mean PANSS-positive: 35.76 in schizophrenia, 39 in BDP) and low CDT scores (median: 5 in both groups) at admission, which may reflect such delays in care, though we did not directly assess this hypothesis. Additionally, hospital conditions during the pandemic—such as reduced staffing, visitor restrictions, or altered treatment protocols—may have impacted the quality and consistency of inpatient care, potentially influencing clinical and cognitive improvements.

As evaluating the indirect effects of the COVID-19 pandemic was not an objective of our study, we did not collect data on variables such as time since last outpatient visit, changes in relapse frequency, or hospital resource constraints. This represents a limitation, as these factors could have contributed to the observed outcomes, particularly the severity of symptoms at admission and the extent of recovery by discharge. We have added this consideration to the revised Discussion under limitations, noting that future studies should explore the pandemic’s indirect effects on psychiatric populations by incorporating variables such as treatment access, relapse history, and hospital conditions during crisis periods. Such research could provide valuable insights into how systemic factors influence recovery trajectories in schizophrenia and BDP.

2. The credentials of the people making the schizophrenia and BDP diagnoses and how the diagnoses were made (e.g., comprehensive evaluation? diagnostic clinical interviewing? by history?) to determine if a participant met diagnostic criteria are not clear.

Thank the reviewer for the valuable comment

“Inclusion criteria were an age of 18 and above, a diagnosis of schizophrenia or BDP according to the Diagnostic and Statistical Manual of Mental Disorders 5 (DSM-5), and consent to participate in the study. The diagnosis was made by a psychiatry attending using a comprehensive psychiatric interview. “

It is added.

3. Participants were assessed at two timepoints (admission and discharge); however, it is not clear how many days participants were hospitalized during that timeframe. It would be helpful to include the average time (and SD) from admission to discharge for each group. I’m wondering if duration of hospital stay could be a factor influencing study results.

Unfortunately, we did not record the duration of the hospital of the patient back then and we could find only 25 patients with their names because we need a national code to access all patients' records. The average stay of those 25 patients is 21.28 days. It is usually the average stay of patients in hospitals in Iran. Patients are usually hospitalized from 14 to 28 days. Given the lack of comprehensive duration data, we could not assess its impact on cognitive (CDT) or clinical (PANSS, CGI) outcomes in full. This represents a limitation of the study, as longer hospital stays could theoretically enhance treatment exposure and influence symptom or cognitive recovery. We have noted this in the revised Discussion under limitations and recommend that future studies systematically record and analyze hospitalization duration as a potential covariate.

4. Authors state that all participants received occupational therapy. How often/how much OT did participants receive? I’m wondering if OT intervention duration plays a role in disorder symptoms and cognitive outcomes.

All participants received occupational therapy (OT) twice weekly throughout their hospitalization, with each session lasting approximately 45 minutes, consistent with standard inpatient psychiatric care protocols at the study sites (Rasoul Akram Hospital and Iran Psychiatry Hospital). As OT was uniformly administered to all patients in both the schizophrenia and BDP groups, it was not included as a variable in the multiple regression analyses. This decision was based on the lack of variability in OT exposure across participants, which precluded its evaluation as a predictor of differential changes in cognitive (CDT) or clinical (PANSS, CGI) outcomes. While OT duration or frequency could theoretically influence symptom severity and cognitive function, the consistent application in this study design limits our ability to assess its specific contribution. Future studies with variable OT dosing could explore its potential moderating effects on these outcomes.

5. It was reported that participants received different biological treatments (i.e., medication and ECT). However, it is unclear how many participants received medication, ECT, or both treatments. What proportion of participants received medication, ECT, or a combination that could alter the frequency/severity of their symptoms and level of cognitive functioning?

We updated the Table 1 for demographic characteristics of the participants

Table 1. Demographic characteristics of the participants

Variable Mean±SD/Median[min-max]/ No.[%] 95% CI Effect size p

Groups(n=84)

Schizophrenia BDP

Age(y) 39[24-65] 40.74±10.82 [0.0035-0.25] 0.02(small) P=0.85

Number of participants 42[%50] 42[%50] - - -

Gender Female: 33[%39.3] 15[%35] 18[%42] - Cohen's w: 0.26(moderate) p < 0.01

Male: 51[%60.7] 27[%65] 24[%58]

ECT 10[%24] 6[%14] - Cohen's w: 0.09 (small) P=0.26

Number of Medication 2[1-4] 2[1-3] [0.13- 0.53] 0.34(moderate) p < 0.01

Mood stabilizer 10[%24] 38[%90] - Cohen's w: 0.64(large) 6.7e-10

Antipsychotic medication

42[%100]

40[%95] - - -

6. Of the

---

## [Decision Letter · Decision Letter 2]

Differential Cognitive and Clinical Improvements in Schizophrenia and Bipolar Disorder Following Hospitalization: A Comparative Analysis Based on the Clock Drawing Test

PONE-D-24-39274R2

Dear Dr. Mirfazeli,

We’re pleased to inform you that your manuscript has been judged scientifically suitable for publication and will be formally accepted for publication once it meets all outstanding technical requirements.

Kind regards,

Thiago P. Fernandes, PhD

Academic Editor

PLOS ONE

Additional Editor Comments (optional):

Reviewers' comments:

Reviewer's Responses to Questions

**Comments to the Author**

1. If the authors have adequately addressed your comments raised in a previous round of review and you feel that this manuscript is now acceptable for publication, you may indicate that here to bypass the “Comments to the Author” section, enter your conflict of interest statement in the “Confidential to Editor” section, and submit your "Accept" recommendation.

Reviewer #3: All comments have been addressed

2. Is the manuscript technically sound, and do the data support the conclusions?

Reviewer #3: Yes

3. Has the statistical analysis been performed appropriately and rigorously? 

Reviewer #3: Yes

4. Have the authors made all data underlying the findings in their manuscript fully available?

Reviewer #3: Yes

5. Is the manuscript presented in an intelligible fashion and written in standard English?

Reviewer #3: Yes

6. Review Comments to the Author

Reviewer #3: I really appreciate the authors' effort and care in addressing all of the reviewer comments. Thank you for the thoroughly re-working sections of the manuscript and for explaining the study data collection process and for full transparency in what data were and were not collected. The only thing that I can think of that may strengthen the study is to add a power analysis (converting F values to Cohen's d) to investigate if the study sample is sufficiently powered to detect within- and between-subject effects.

7. PLOS authors have the option to publish the peer review history of their article (what does this mean? ). If published, this will include your full peer review and any attached files.

**Do you want your identity to be public for this peer review?** For information about this choice, including consent withdrawal, please see our Privacy Policy .

Reviewer #3: No

---

## [Editor Report · Acceptance letter]

PONE-D-24-39274R2

PLOS ONE

Dear Dr. Mirfazeli,

I'm pleased to inform you that your manuscript has been deemed suitable for publication in PLOS ONE. Congratulations! Your manuscript is now being handed over to our production team.

Kind regards,

on behalf of

Dr. Thiago P. Fernandes

Academic Editor

PLOS ONE